# Effects of Outlet Shrinkage on Hydraulics in Hyper-Concentrated Sediment-Laden Flow

**Jijian Lian [1,2], Hongxia Yin [1,2], Fang Liu [1] , Huiping Li [1] and Wenjuan Gou [1,*]**

[1]   State Key Laboratory of Hydraulic Engineering Simulation and Safety, Tianjin University, Tianjin 300072, China; jjlian@tju.edu.cn (J.L.); yinhongxia@tju.edu.cn (H.Y.); fangliu@tju.edu.cn (F.L.); lihuiping@tju.edu.cn (H.L.)

[2]   School of Water Conservancy and Hydroelectric Power, Heibei University of Engineering, Handan 056038, Hebei, China

*   Correspondence: gwj@tju.edu.cn; Tel.: +86-022-27401137

**Abstract:** Finding an appropriate shape for the releasing building is thoroughly relevant given the energy dissipation and safety requirements of a high dam in a sediment-laden river. Thirty-six physical experiments on trajectory energy dissipation were conducted, researching the influence of three overflow shapes (contraction ratios of 0.5, 0.4, and 0.3) with four sediment concentrations (0, 50, 150, and 250 kg/m$^3$) on the discharge, flow regime, and hydrodynamic pressure of a plunge pool slab. The experimental results demonstrated that the flow coefficient gradually decreased as the contraction ratio decreased in a relatively high weir head, regardless of the sediment concentration. The water nappe narrowed and the length of the longitudinal trajectory increased as the outlet shrinkage and sediment concentration decreased. With the increase in sediment concentration, the nappe regime approached stability, and the flow in the plunge pool tended toward small rolling, causing the impact pressure and fluctuating pressure to increase. Changes in overflow shape had little effect on the position of pressure peak, but the value became lower as the ratio diminished. The influence on the hydrodynamic pressure by outlet shrinkage became attenuated while the sediment concentration increased. The fluctuating energy and vortex scale were enhanced due to the increased viscosity with increasing sediment concentrations.

**Keywords:** sediment-laden flow; outlet shrinkage; flow regime; jet; fluctuating pressure

## 1. Introduction

Since the mid-20th century, the construction of large-scale dams around the world has increased significantly. Water storage and power generation were increased significantly by increasing reservoir water levels and dam height. The increase in reservoir water level was accompanied by huge energy increases. Energy dissipation is an important issue in the design of a hydraulic structure, especially when dam parameters include a high water head and large discharge. If this energy was not dissipated well, the safety of the side slopes and downstream plunge pools would be threatened. The form of energy dissipation to be used in hydraulic engineering was an urgent design issue. At present, two types of energy dissipation are used. Some are traditional energy dissipation methods (hydraulic jump energy dissipation, surface flow energy dissipation, and ski-jump energy dissipation); others are new energy dissipation methods (such as flaring gate pier, slit-type, stepped energy dissipation, or a combination of various energy dissipations) [1,2]. According to the rough statistics of existing projects, the projected parts cost for flood discharge and energy dissipation accounted for 40%–50% of the total project investment. Thus, shape optimization for the overflowing building was important for the safety of the large hydro-junction and the investment of the project [1–3].

An impinging jet formed by the water nappe directly impacting a solid wall has been encountered in many engineering fields. In hydropower projects, the water nappe ejected from the overflow dam can also be regarded as an impinging jet in the water entry stage. The rapid spread of the jet in the pool dissipates the energy of the dropped water and weakens the fluctuations at the bottom of the pool [1]. The research results showed that the destructive effect of the impinging jet was not only related to the mean value of impact pressure, but also to the characteristics of the fluctuating pressure [2,3].

The pressure fluctuation characteristics were affected by the turbulent structure. Methods of turbulence characteristics include theory, experimentation, and simulation, such as direct numerical simulation (DNS) [4], and large eddy simulation (LES) [5]. Gutmark et al. [6–8] proposed that energy dissipation depends on the turbulence of flow and the fluctuation of turbulence based on the formation of a coherent structure. Chakraborty et al. [9–11] studied the scale, intensity, and density of vortex structures. The popular vortex identification criteria and structure recognition methods were also analyzed. Some advanced experimental equipment, such as particle image velocimetry [12], has also been used in research.

Some large hydro-junctions were built on hyper-concentrated sediment-laden rivers [13]. The rheological properties of the water flow [14,15] changed due to the large quantity of viscous particles. With a sediment-laden water flow, Wang and Qian [16] studied the turbulent structure of the open channel flow. Their results showed that the fundamental turbulent structure of the Newtonian fluid containing sediment remains unchanged, but while the turbulence intensity is smaller than in a freshwater flow, the vortex of turbulence is larger; however, Wang et al. [17] found different experimental results. Guo and Julien [18] conducted a theoretical analysis to determine that sediment suspension increased mainstream energy loss and reduced vertical turbulent diffusion and increased velocity gradient. Samanta et al. [19] used DNS to simulate porous turbulence and observed a decrease in the flow turbulence intensity in the porous conduit.

The hydraulic characteristics of the impact jet and the energy dissipation in the plunge pool have been extensively studied in fresh water [20]. Gou et al. [21] examined the energy dissipation of the sediment-laden water flow in a ski jump, and found that the flow discharge is similar to that of fresh water; the turbulent vortex becomes larger and the turbulent energy is reduced; the fluctuating energy and vortex scale both increase with the increasing sediment concentration due to the increased viscosity.

The flaring gate pier widens the tail part of the traditional flat tail pier, so that the water flow forms a narrow and long flow regime at the outlet of the overflow, increasing dispersion and air entrainment of the water nappe in the longitudinal direction and improving the energy dissipation efficiency. As early as the 1970s, Lin [22] proposed the idea of setting a flaring gate pier on the top of the arch dam to make the water nappe fully disperse in the longitudinal direction and promote energy dissipation by air entrainment. In 1974, Gong [23] proved that energy dissipation can be significantly improved in hydraulic model tests with flaring gate piers. In recent years, flaring gate piers of various types have been used in many projects, such as Y-shaped, X-shaped, new X-shaped, and T-shaped flaring gate piers [24].

In the design for energy dissipation, the change in overflow shape was also affected by sediment concentration. However, the research on shape optimization of overflow was almost all about fresh water. The flaring gate pier was a common optimization for energy dissipation.

In earlier research in the laboratory, the effect of sediment concentration on the deep outlet was noted [21]. In this paper, the effect of the shape of surface outlet was considered. Model experiments were conducted using a recirculating flume with three overflow shapes (contraction ratio: 0.5, 0.4, 0.3) at four sediment concentrations ($0 \text{ kg/m}^3$, $50 \text{ kg/m}^3$, $150 \text{ kg/m}^3$, and $250 \text{ kg/m}^3$). The influence of sediment concentrations and outlet shrinkages on the water nappe were studied, along with the flow regime and pressure fluctuations in the plunge pool.

## 2. Materials and Methods

### *2.1. Experimental Setup*

#### 2.1.1. Model Design

A physical model of the overflow structure was designed for a recirculating tank with a plunge pool, as shown in Figure 1. The model consisted primarily of an upstream tank with a water level indicator that supported a stable head, a plunge pool, and a mud pump with a flow meter. The upstream water tank was 1 m wide, 1.3 m long, and 2.3 m high. The plunge pool was 0.65 m deep and 4 m long with a trapezoidal cross section with 0.8 m and 0.4 m width at the top and bottom, respectively. The upstream tank and plunge pool were connected by a 12 cm diameter pipe and mud pump. An overflow weir was fixed on the side wall of the upstream tank, 2 m away from the bottom of the plunge pool. The outlet sections of the overflow with three contraction ratios are shown in Figure 2. The contraction ratio ε is defined as the ratio of the width of the outlet and the inlet and the smaller contraction ratio of overflow conduced the narrower outlet width.

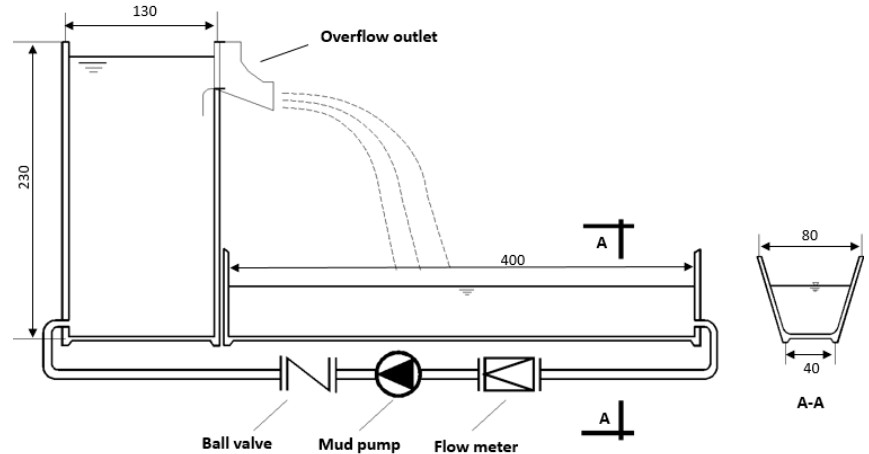

**Figure 1.** Model arrangement (Unit = cm).

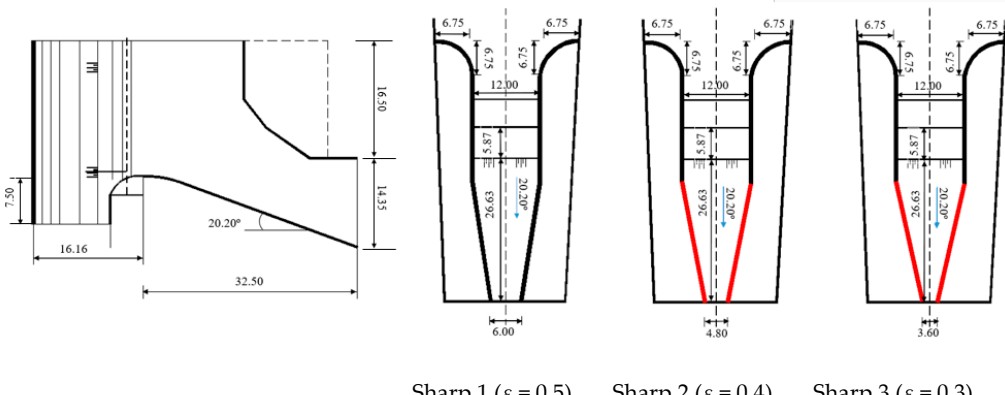

Sharp 1 (ε = 0.5)    Sharp 2 (ε = 0.4)    Sharp 3 (ε = 0.3)

**Figure 2.** Dimension of overflow surface with different contraction ratios (Unit = cm).

Experiments were carried out in a room-temperature environment of about 20 ± 2 °C. The upstream and downstream water heads were controlled within ±1 mm experimental error. Fifteen measuring points were arranged along the middle line on the bottom of the plunge pool, as shown in Figure 3. Measuring point #1 was 50 cm away from the beginning of the plunge pool, that is, the end of the outlet of overflow, and the interval of measuring points was 5 cm in the impact region (points #2–12), and the interval of measuring points was 10 cm in the rest position (points #1–2 and #12–15).

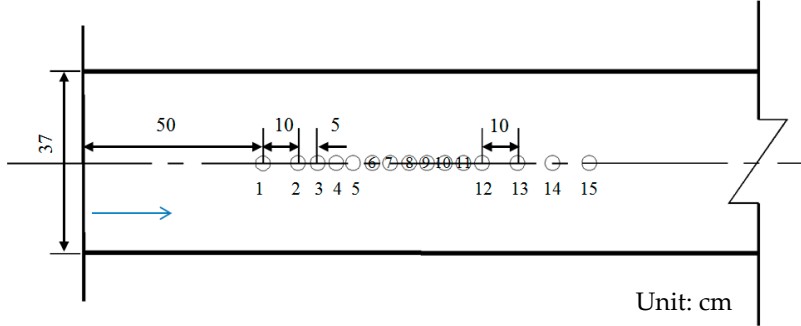

**Figure 3.** Test point arrangement.

Experiments were conducted by using diaphragm-type TS202 pressure transducers (Beijing Tiantai Xingye Technology Co., Ltd., Beijing, China) with a bearing surface diameter of 2 cm. The data were collected by the DASP (Data Acquisition and Signal Processing) data collection system with a sampling frequency of 100 Hz. The inner and outer edges of the water nappe were measured by a tape. The viscosity of the hyper-concentrated sediment-laden flow was measured by a rotational viscometer NDJ-8S (Shanghai Yueping Scientific Instrument Co., Ltd., Shanghai, China). The sediment was fine silt and clay, with a particle size ranging from 0.001 mm to 0.05 mm, a median diameter of 0.008 mm, and a bulk density of 2.4 kg/m$^3$. In order to ensure accurate experimental results, the sediment concentration was monitored and adjusted and the transducers were kept uncovered by sediment at all times to ensure correct test results. The results were averaged from the data of multiple measurements. The pressure transducer was calibrated without water and the hydrostatic pressure was measured prior to the experiment to provide a comparison of the water levels.

### 2.1.2. Experimental Conditions

The influence of outlet shrinkage on the hydraulic characteristics of the sediment-laden flow was studied. A total of 36 experiments were carried out, with the specific experimental conditions shown in Table 1. The experiment used four sediment concentrations of sediment-laden flow including fresh water (0 kg/m$^3$, 50 kg/m$^3$, 150 kg/m$^3$, and 250 kg/m$^3$) and three contraction ratios (0.3, 0.4, and 0.5) of the overflow weir. The upstream water heads were 2.20 m, 2.15 m, and 2.10 m and the downstream water heads were 0.15 m. The experimental error was controlled below 1%. The viscosity of the sediment-laden fluid was 5 mPa·s at a concentration of 50 kg/m$^3$, 12 mPa·s at a concentration of 150 kg/m$^3$, and 30 mPa·s at a concentration of 250 kg/m$^3$.

**Table 1.** Experimental conditions.

| | Shape 1 | | | | Shape 2 | | | | Shape 3 | | |
|---|---|---|---|---|---|---|---|---|---|---|---|
| No. | Contraction Ratio | Upstream Water Level (m) | Sediment Concentration (kg/m³) | No. | Contraction Ratio | Upstream Water Level (m) | Sediment Concentration (kg/m³) | No. | Contraction Ratio | Upstream Water Level (m) | Sediment Concentration (kg/m³) |
| 1 | 0.5 | 2.20 | 0 | 13 | 0.4 | 2.20 | 0 | 25 | 0.3 | 2.20 | 0 |
| 2 | 0.5 | 2.15 | 0 | 14 | 0.4 | 2.15 | 0 | 26 | 0.3 | 2.15 | 0 |
| 3 | 0.5 | 2.10 | 0 | 15 | 0.4 | 2.10 | 0 | 27 | 0.3 | 2.10 | 0 |
| 4 | 0.5 | 2.20 | 50 | 16 | 0.4 | 2.20 | 50 | 28 | 0.3 | 2.20 | 50 |
| 5 | 0.5 | 2.15 | 50 | 17 | 0.4 | 2.15 | 50 | 29 | 0.3 | 2.15 | 50 |
| 6 | 0.5 | 2.10 | 50 | 18 | 0.4 | 2.10 | 50 | 30 | 0.3 | 2.10 | 50 |
| 7 | 0.5 | 2.20 | 150 | 19 | 0.4 | 2.20 | 150 | 31 | 0.3 | 2.20 | 150 |
| 8 | 0.5 | 2.15 | 150 | 20 | 0.4 | 2.15 | 150 | 32 | 0.3 | 2.15 | 150 |
| 9 | 0.5 | 2.10 | 150 | 21 | 0.4 | 2.10 | 150 | 33 | 0.3 | 2.10 | 150 |
| 10 | 0.5 | 2.20 | 250 | 22 | 0.4 | 2.20 | 250 | 34 | 0.3 | 2.20 | 250 |
| 11 | 0.5 | 2.15 | 250 | 23 | 0.4 | 2.15 | 250 | 35 | 0.3 | 2.15 | 250 |
| 12 | 0.5 | 2.10 | 250 | 24 | 0.4 | 2.10 | 250 | 36 | 0.3 | 2.10 | 250 |

*2.2. Methods*

### 2.2.1. Nyquist's Law

Nyquist's Law [25] states that a signal must be sampled at at least twice its highest analog frequency in order to extract all of the information from the bandwidth. Sampling at slightly more than twice the frequency will make up for imprecisions in filters and other components used for the conversion. The choice of sampling interval and sampling time was related to the ability of the acquired data to accurately reflect the experimental reality. According to the results of previous research [24], a sampling frequency of 100 Hz was used to satisfy Nyquist's law [25].

According to the sampling theorem, the truncation frequency must be greater than the highest frequency of the sampled signal in the selection. If the sampling interval is too large, the truncation frequency would be too small, resulting in spectral aliasing. Therefore, at least 8192 samples were collected per experiment to provide sufficient data to reduce uncertainty. The valid data for the complete sample set were analyzed and were used to calculate the parameters of the hydrodynamic pressure in MATLAB (The MathWorks, Inc., Natick, MA, USA).

### 2.2.2. Probability Density

It is known from probability theory and mathematical statistics theory that, for a random variable, $x$, with a mean of zero, the probability density is given by

$$p(t) = \frac{1}{\sqrt{2\pi}\sigma} e^{-\frac{x^2}{2\sigma^2}}. \tag{1}$$

The skewness coefficient is given by

$$C_s = \frac{1}{N} \frac{\sum\limits_{i=1}^{N} x^3}{\sigma^3} \tag{2}$$

and the kurtosis coefficient is given by

$$C_E = \frac{1}{N} \frac{\sum\limits_{i=1}^{N} x^4}{\sigma^4}, \tag{3}$$

where $N$ is the number of variables and $\sigma$ is the root mean square of variables.

If $C_S > 0$, the value of fluctuating pressure has a greater probability to be positive, which is called positive deviation. On the contrary, negative deviation is $C_S < 0$. The kurtosis coefficient is also called the flatness factor. Some intermittent random variables have a large absolute fluctuation in the time series. When the measured variable is intermittent, $C_E > 3$, the probability density function is high and thin.

### 2.2.3. Power Spectrum and Correlation Coefficient

One of the most important methods in signal processing is the Fourier Transform, which bridges the time and frequency domains. Suppose the random process $f(t)$, whose Fourier transform is

$$F(f) = \int_{-\infty}^{\infty} f(t)e^{-i\omega t}dt \tag{4}$$

The inverse transformation is

$$f(t) = \frac{1}{2\pi} \int_{-\infty}^{\infty} F(f)e^{-i\omega t}d\omega \tag{5}$$

The Fourier transform of the autocorrelation function is defined as the power spectral density function:

$$S(f) = 4 \int_0^\infty R(t) \cos 2\pi f t \, dt, \tag{6}$$

where $f$ is the frequency of the signals, $t$ is the test time, and $S(f)$ is the power spectrum of the signals.

The spatiotemporal correlation coefficient of the signals is expressed as follows:

$$\rho(x, l, \tau) = \frac{\overline{p'(x,t)p'(x+l,t+\tau)}}{\left[\overline{p'^2(x,t)p'^2(x+l,t+\tau)}\right]^{1/2}}, \tag{7}$$

where $\tau$ is the time delay; $l$ is the distance between the two measuring points; $p'(x,t)$ is the measured value of point $x$ at moment $t$; $p'(x+l,t+\tau)$ is the measured value of point $(x+l)$ at $(t+\tau)$ moment; $\overline{p'(x,t)p'(x+l,t+\tau)}$ is the average value of the measured value at two points $x$ and $(x+l)$; $\rho(x,\tau)$ is the autocorrelation coefficient at $l = 0$, which reflects the correlation of the same measuring point at different times; and $\rho(x,l)$ is the instantaneous spatial correlation coefficient when $\tau$ equals 0.

The power spectrum is an important parameter representing the frequency domain characteristics of fluctuating pressure. The instantaneous spatial correlation coefficient reflects the magnitude of the vortex feature scale.

## 3. Experimental Results

### 3.1. Flow Discharge

The discharge of sediment-laden flow is a key control factor in the design of hydraulic structures in a hyper-concentrated river, and an appropriate flow rate ensures the safety of dams. Figure 4 depicts the relationships of discharges with two concentrations of sediments (0, 150 kg/m$^3$) for three overflow shapes. The experimental results indicated that the discharge capacity of sediment-laden flow was basically similar to that of fresh water. The discharge capacity was only related to the upstream water head, regardless of the sediment concentration; this is consistent with the trends of other studies [26–28]. Therefore, it was only necessary to evaluate the flow rate based on fresh water when designing a hydraulic structure.

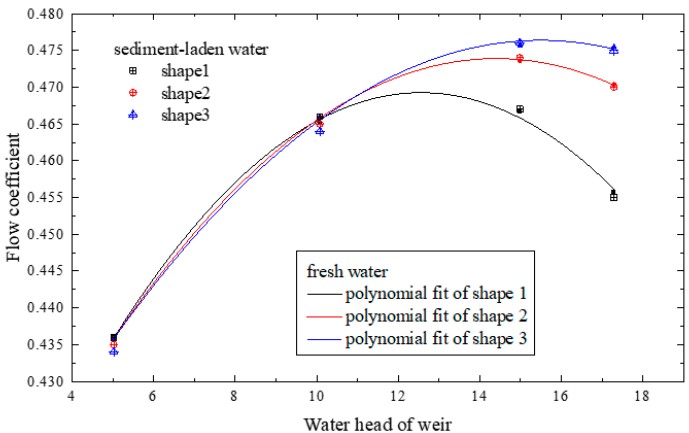

**Figure 4.** Relationship between the upstream water head and the flow coefficient with different contraction ratios for fresh water and sediment-laden flow with concentrations of 150 kg/m$^3$.

The outlet shrinkage of the overflow weir definitely affects the discharge capacity. The discharge capacity of the weir can be expressed by the flow coefficient as

$$m = \frac{Q}{b\sqrt{2g}H^{3/2}},$$

(8)

where *m* is the flow coefficient of the overflow; *Q* is the discharge capacity; *b* is the overflow weir width; and *H* is the water head of the weir. The relationship between the flow coefficient of the weir and the upstream head for the three contraction ratios is shown in Figure 4. The results indicated that the flow coefficient of different shape of overflow weirs generally increased first and then decreased with increasing water head. When the water head was below 10 cm, the coefficient merely increased with the water head and was not related to the contraction ratios—that is, the discharge capacity of the overflow weir was not affected by the width of the outlet. When the water head of the weir was over 10 cm, the flow coefficient gradually decreased with a decreasing contraction ratio. The influence of outlet shrinkage on the discharge capacity of the weir gradually increased. As the decreased contraction ratio gradually reduced the width of the outlet and enlarged the water depth, this affected the discharge capacity of water flow.

### 3.2. Flow Regime

### 3.2.1. Water Nappe in Air

The changes of overflow shape had an effect on the nappe surface and width. The flow regimes of water nappe are shown in Figure 5. Figure 5a–c depicts the water nappe in fresh water with contraction ratios of 0.3, 0.4, and 0.5 at an upstream water head of 2.20 m and a downstream water head of 0.15 m. The water nappe was smooth at the outlet and the air entrainment was obvious in the nappe as the water nappe dropped. The water nappe became deep due to the narrow width of the outlet. This caused air entrainment and dispersion of the water nappe. The air entrainment phenomenon of shapes 1 and 2 was not obvious, but in shape 3 it was obvious. The aerated part was mainly concentrated in the middle and lower part of water nappe. Figure 5d–f depicts the water nappe of sediment-laden flow with a sediment concentration of 250 kg/m$^3$ at an upstream water head of 2.20 m and downstream water head of 0.15 m. The results show that the nappe of sediment-laden flow in the outlet was similar to that of fresh water, the water nappe of sediment-laden flow was smoother than fresh water, and small waves appeared slightly in the lower part.

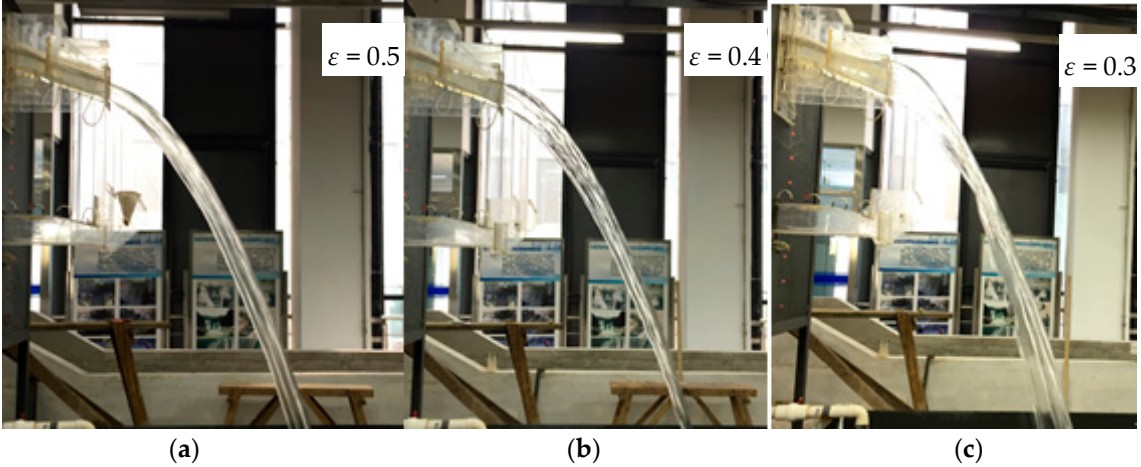

(a)　　　　　　　　　　　　(b)　　　　　　　　　　　　(c)

**Figure 5.** *Cont.*

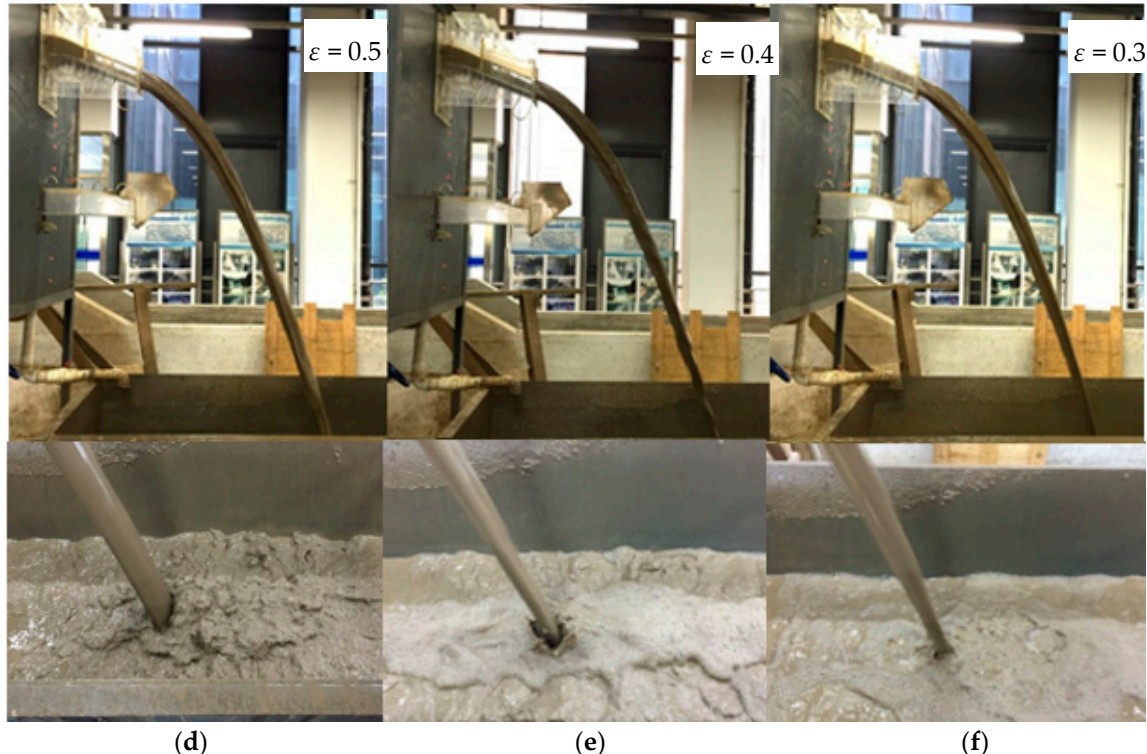

**Figure 5.** Water nappe regime at upstream water head of 2.20 m and downstream water head of 0.15 m: (**a**) fresh water at $\varepsilon$ of 0.5; (**b**) fresh water at $\varepsilon$ of 0.4; (**c**) fresh water at $\varepsilon$ of 0.3; (**d**) sediment-laden flow with a concentration of 250 kg/m$^3$ at $\varepsilon$ of 0.5; (**e**) sediment-laden flow with a concentration of 250 kg/m$^3$ at $\varepsilon$ of 0.4; (**f**) sediment-laden flow with a concentration of 250 kg/m$^3$ at $\varepsilon$ of 0.3.

The water nappe became narrower with outlet shrinkage and as air entrainment increased. However, the sediment concentration caused attenuated air entrainment and dispersion of the nappe due to the viscosity of the fluid [21,29]. The existence of sediment in the flow was beneficial to stabilize the regime of the water nappe.

3.2.2. Jet Trajectory

The length of the jet trajectory was analyzed for different contraction ratios, as shown in Figure 6. The experimental results indicate that the trajectory increased with a decrease in contraction ratio. The outlet of shape 3 was narrower and the contraction ratio decided the flow regime of the water nappe. The enhanced air entrainment enlarged the trajectory. Therefore, the trajectories of shape 1 and 2 were two parallel lines, while the trajectory of shape 3 sharply increased.

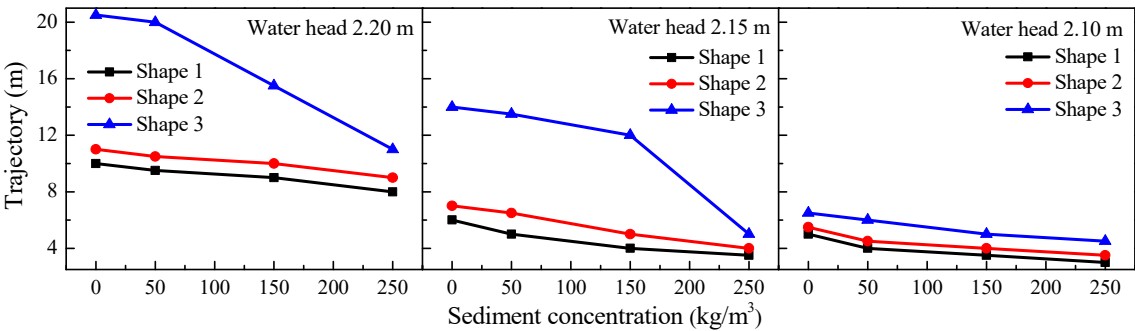

**Figure 6.** Relationship between lengths of jet trajectory and sediment concentrations for three shapes of overflow at three water heads.

In fresh water, the jet trajectory was mainly affected by factors such as the shape of overflow and the upstream and downstream water head. With an increase of sediment concentration, the length of the jet trajectory decreased. The length was slightly reduced for shapes 1 and 2. The water nappe had slight air entrainment and the influence of sediment concentration was relatively small. For shape 3, though using a narrower outlet, the water nappe was still heavily aerated at a low-sediment concentration, the length of the jet trajectory decreased slightly, and the contracted water nappe diffused significantly along the longitudinal direction, obviously reducing the kinetic energy carried in the water. However, while the sediment concentration increased to a critical level, the length of jet trajectory decreased abruptly and sharply. The critical concentration was relative to the upper water head.

The water nappe exhibited different flow regimes with different sediment concentrations. In general, the distance between the inner and outer edges (i.e., the length of the jet trajectory) decreased with the increase of the sediment concentration. The length change of the jet trajectory was affected by air entrainment and gravity, but the increase of viscosity reduced the air entrainment of the water jet in the air. This phenomenon was attributed to the higher viscosity of the sediment-laden flow [21,29]. Additionally, the influence of the outlet shrinkage on the water nappe became smaller. This was due to the increase in effective gravity of the hyper-concentrated sediment-laden flow. Velocity component in the direction of gravity of the water nappe with higher sediment concentrations was greater than that with lower sediment concentrations.

3.2.3. Flow Regime in Plunge Pool

When flooding with a ski-jump type energy dissipator, the water nappe fell from the overflow of the dam into the plunge pool to form an impinging jet. In order to improve the stability of the water cushion floor in projects, jet diffusion law, flow rate attenuation law, and the impact pressure of water nappe on the bottom received more attention. Hartung et al. [30] first began to study the diffusion law of jetted water in the plunge pool through model tests and a schematic diagram of the flooding impinging jet model defined commonly in engineering [31], as shown in Figure 7. Normally, the bottom can be divided into three regions according to the difference in surface hydrodynamic pressure distribution. Region I was the water nappe impact region, which directly bears the influence of the jet impact and the surface pressure, and was much larger on the bottom of the pool. Region II was the wall jet region, where the jet was constrained by the side wall of the pool, the streamline was curved, and the surface pressure was sharply reduced. The uplift force of water flow was relatively strong at Region II and the bottom plate was unstable. Region II' corresponded to Region II, where the hydrodynamic pressure was similar to that of Region II. Region III was an area of gradual change in water flow.

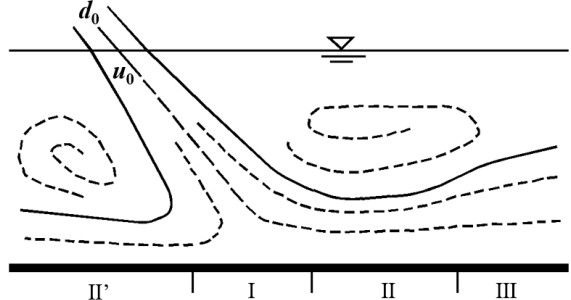

**Figure 7.** Schematic diagram of an impinging jet model [31].

The flow regime in the pool is shown in Figure 8 for different outlet shapes with various sediment concentrations. As shown in Figure 8a, the water nappe carried a lot of kinetic energy impinged on the water surface of the plunge pool and sharp rolling formed in fresh water with shape 1. The water surface fluctuated strongly and a large amount of milky white bubbles and water splashes were obvious. With shape 3 in fresh water, the water nappe induced a large amount of air entrainment,

and the kinetic energy was obviously reduced. The rolling formed in the plunge pool was obviously reduced compared with shape 1, and the water surface fluctuation was relatively steady. When the water flow was sediment-laden, the viscosity of water in the plunge pool sharply increased, there were not many splashes, and the number of small-scale vortices was obviously reduced. Therefore, the water flow of shape 3 was obviously weakened. The flow regime was affected by jet characteristics, as the break-up and air-entrainment of the jet were affected by surface tension and turbulence [11]. The viscosity of the water flow increased with the presence of sediment, producing a gravity effect and reducing the air entrainment [21].

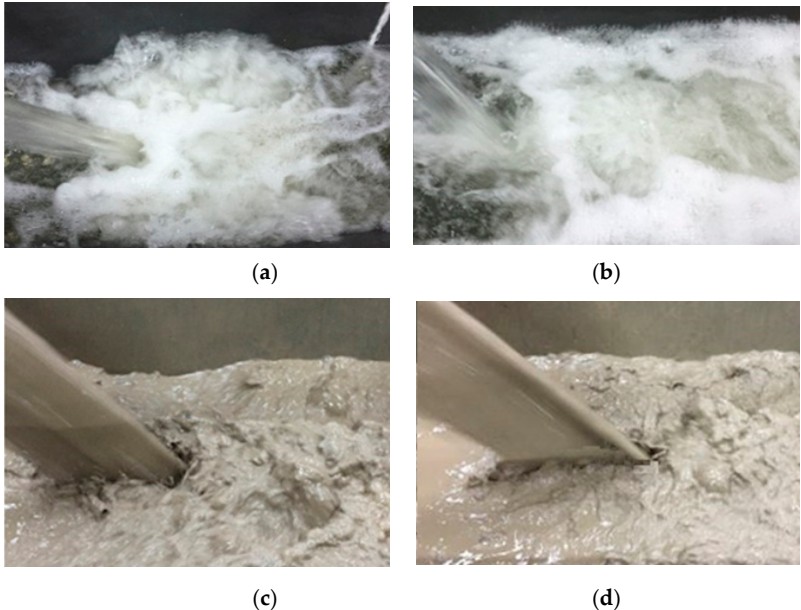

**Figure 8.** Flow regime of impact region in the plunge pool with upstream water head of 2.20 m and downstream water head of 0.15 m; (**a**) shape 1 of fresh water; (**b**) shape 3 of fresh water; (**c**) shape 1 of sediment-laden flow with a concentration of 150 kg/m$^3$; (**d**) shape 3 of sediment-laden flow with a concentration of 150 kg/m$^3$.

### 3.3. Hydrodynamic Pressure

The hydrodynamic pressure of the plunge pool is an important index to evaluate the stability of the protection structure and the effect of energy dissipation of the high dam. This was affected by the diffusion of the jet and the scale of turbulence in the plunge pool. The instantaneous value of pressure can be expressed as a combination of mean pressure and fluctuating pressure at a measuring point of the plunge pool, given as

$$P = \overline{P} + P',\tag{9}$$

where $P$ is the instantaneous pressure, $\overline{P}$ is the mean pressure, and $P'$ is the fluctuating pressure. The average value of hydrodynamic pressure of the impinging jet on the bottom of the plunge pool can be determined by the average pressure over time, so the relative time-averaged pressure ($P^*$) can be given as

$$P^* = \overline{P} - h_0,\tag{10}$$

where $h_0$ is the static water depth in the plunge pool. The relative time-averaged pressure is a function of the position of the experimental measurement point. The root mean square is a statistical characteristic value indicating that the random variable deviates from the average value, and the root mean square of the fluctuating pressure can be used to indicate the intensity of water flow turbulence. The calculation formula is

$$\sigma_P = \sqrt{P'^2} = \sqrt{\left(P - \overline{P}\right)^2}, \tag{11}$$

where $\sigma_P$ is the root mean square of the fluctuating pressure in the plunge pool.

### 3.3.1. Relative Time-Averaged Pressure

The fluctuating pressure of the bottom plate was measured synchronously. The relative time-averaged pressure distribution along the centerline of the bottom is shown in Figure 9a–c, where the ordinate is the relative time-averaged pressure at each test point, and the abscissa is the distance between the measuring point and the beginning of the plunge pool.

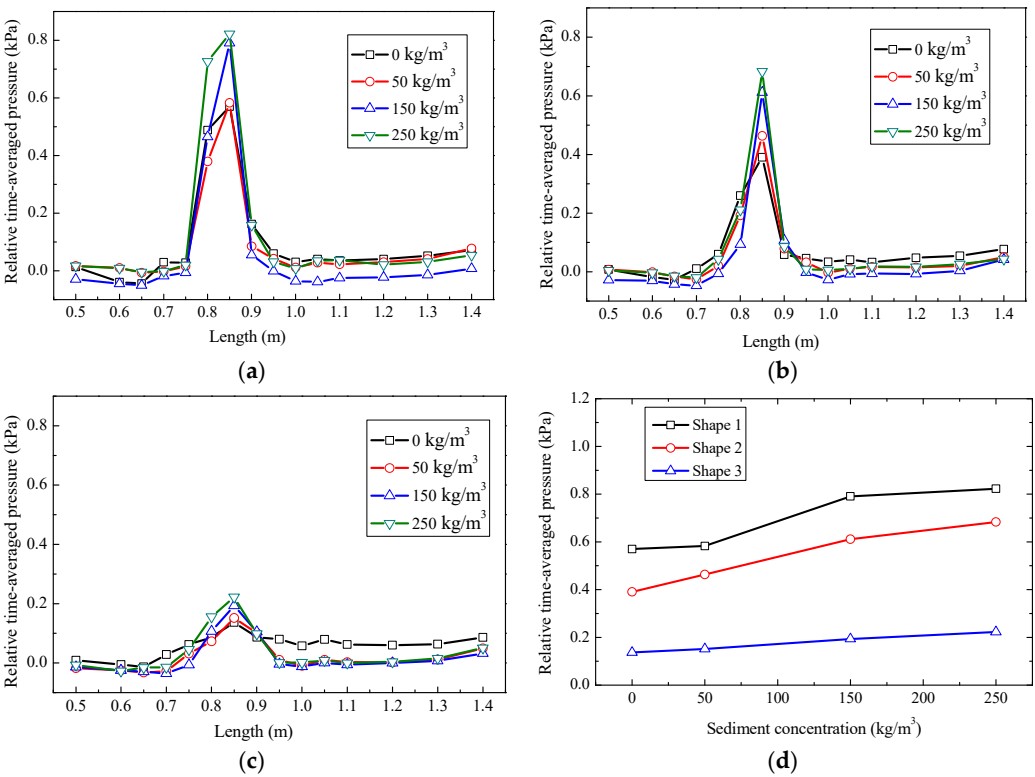

**Figure 9.** Relationship between the relative time-averaged pressure and sediment concentrations at various shapes; (**a**) shape 1; (**b**) shape 2; (**c**) shape 3; (**d**) pressure comparison for various sediment concentrations at the impact point.

The maximum value of the relative time-averaged pressure affected the stability and safety of the bottom of the plunge pool. Observing the value of time-averaged pressure, the results indicated that the distribution of pressure for the hyper-concentrated sediment-laden flow was consistent with that of fresh water. The relative time-averaged pressure distribution in the front of the plunge pool, away from the jet impact region, was relatively stable, and its value was basically proportional to the water depth. The relative time-averaged pressure in the jet impact region increased sharply, forming a distinct peak. The position where the peak appeared was only relevant to the upstream and downstream water head, regardless of the alteration in sediment concentration; with the same upstream and downstream water head, the peak value of relative time-averaged pressure increased continuously with the increase in sediment concentration, consistent with the experimental results of Gou [21]. The time-averaged pressure away from the water nappe impact region tended to be stable at first and then rose slightly, as the nappe fell into the plunge pool and formed a dammed water area downstream of the impact region. The water depth in this area was greater than the plunge pool, so the time-averaged pressure increased again.

Figure 9d depicts the relative time-averaged pressure at the impact point. The result shows that the narrower outlet width caused more air entrainment of the water nappe and a lower impact pressure on the bottom. As the sediment concentration increased, the dispersion of the water nappe decreased and the effect of shape changes of the outlet was reduced. As the sediment concentration increased, the diffusion and air entrainment of the jet decreased. Higher pressure occurred due to the increase in viscosity of the liquid.

### 3.3.2. Fluctuating Pressure

The average hydrodynamic pressure indicated the average impact ejection force on the bottom of the plunge pool over a period of time and the fluctuating pressure caused by turbulence indicated the degree of turbulence at each test point. Figure 10a–c depicts the relationship between the root mean square (RMS) of fluctuating pressure along the center line of pool and the distance of the measuring point far from the beginning of the plunge pool.

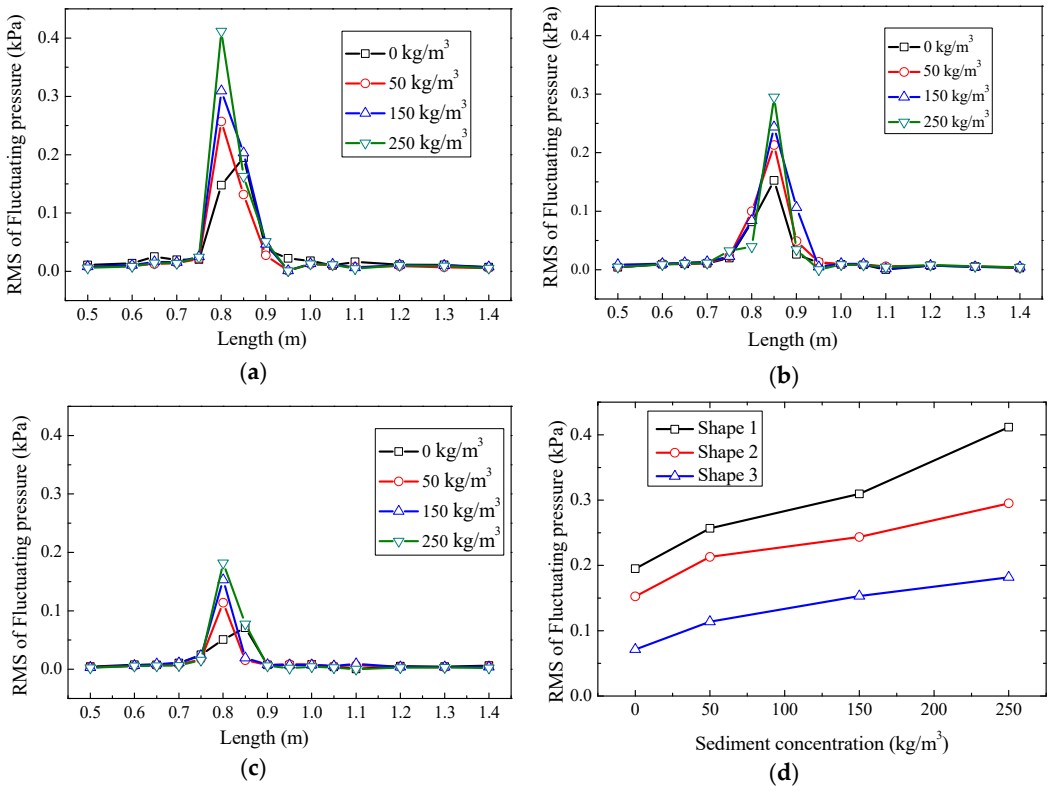

**Figure 10.** Distribution of fluctuating pressure at three shapes for different concentrations at upstream water head of 2.20 m and downstream water head of 0.15 m; (**a**) shape 1; (**b**) shape 2; (**c**) shape 3; (**d**) fluctuating pressure of impact point.

The results indicated that the RMS of fluctuating pressure remained stable at the beginning of the plunge pool and then increased sharply to a peak value, before eventually returning to a stable value. For hyper-concentrated sediment-laden flows with a given upstream water level, the maximum value of RMS increased as the sediment concentration increased from 0 kg/m$^3$ to 250 kg/m$^3$. An increase in sediment concentration led to an increase in flow viscosity, resulting in a decrease in turbulence intensity [32] and an increase in turbulent fluctuation, which was consistent with the turbulence characteristics of sediment flow in open channels [33]. The shape of the outlet had a great influence on the pressure. As the outlet width narrowed, the maximum RMS of fluctuating pressure at the impact point reduced, and the length of the region after the peak decreased. This indicated that the narrower

outlet width caused more air entrainment and more energy dissipation in air and less impact pressure. The change in shape can also slightly shift the impact point further down.

Figure 10d depicts the trends for the maximum RMS at four sediment concentrations with three shapes. The experimental results illustrated that the change trend of maximum fluctuating pressure in three shapes and the narrower outlet resulted in a smaller fluctuating pressure. The impinging jet produced the amount of vortex in the pool and the movement and interaction of vortices with different scales in the plunge pool caused the fluctuating pressure. The hyper-concentration in liquid results in a higher viscosity. This resulted in a smaller-scale vortex reduction and a larger-scale vortex increase, resulting in an increase of the fluctuating pressure, and also an increase of the dissipation energy.

## 4. Discussion

### 4.1. Probability Density of Fluctuating Pressure

The probability density is an important statistical parameter that describes the fluctuating pressure and represents its overall amplitude. The kurtosis coefficient $C_S$ (Equation (2)) and skewness coefficient $C_E$ (Equation (3)) of fluctuating pressure characterize its normality well and are key features with which to further study the maximum fluctuating pressure.

Figure 11 depicts the probability density of the maximum fluctuating pressure for different overflow weirs at different sediment concentrations at the 2.20 m upstream water level. The abscissa was the standard amplitude normalized by the maximum fluctuating pressure ($(p-\mu)/\sigma$, where $p$ is the instantaneous value of the maximum fluctuating pressure, $\mu$ is the average value of the maximum fluctuating pressure, $\sigma$ is the root mean square of the maximum fluctuating pressure), and the ordinate is the probability density. The probability density of the maximum pressure basically satisfied the normal distribution. As the sediment concentration continued to increase, the influence of shape change on the probability density gradually reduced.

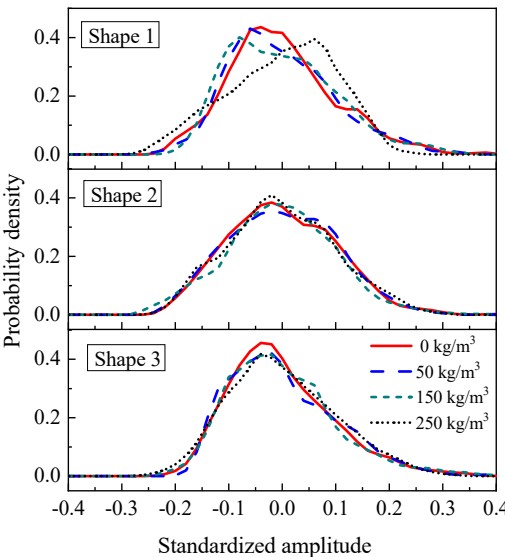

**Figure 11.** Profile of probability density of the maximum fluctuating pressure for three shapes at various sediment concentrations in 2.20 m upstream water levels.

Table 2 shows the skewness coefficient $C_S$ and kurtosis coefficient $C_E$ of the fluctuating pressure at the typical measuring point at the bottom while the water head is 2.20 m. The skewness coefficient of the fluctuating pressure was concentrated from 0 and 1 and indicated that the probability density of fluctuating pressure appeared for positive skew; this result corresponds well with that in [21]. The kurtosis coefficient mostly exceeded 3, and the probability density of the fluctuating pressure on the bottom was higher and thinner compared with the standard normal distribution. The results

reflected that, in the swirling area of flow, the fluctuating pressure deviates from the normal distribution due to the influence of swirling and a vortex. The strict normal distribution requires that fluctuating pressure is unbounded, affecting factors must be independent of each other, and there can be no obviously dominant item. In fact, the fluctuating pressure is a random load under the action of water flow. At certain locations, due to the influence of the flow regime and boundary conditions, there are always dominant factors in the flow. Therefore, the fluctuating pressure cannot be a strictly normal distribution [34].

**Table 2.** Skewness coefficient $C_S$ and kurtosis coefficient $C_E$ at 2.20 m water head.

| Sediment Concentration (kg/m$^3$) | $C_S$ | | | | $C_E$ | | | |
|---|---|---|---|---|---|---|---|---|
| | **0** | **50** | **150** | **250** | **0** | **50** | **150** | **250** |
| Shape 1 | 0.5756 | 0.7667 | 0.6073 | 0.2048 | 3.4021 | 3.6736 | 2.9954 | 2.8885 |
| Shape 2 | 0.3059 | 0.1312 | 0.0717 | 0.2021 | 2.9309 | 3.5226 | 3.2167 | 2.6378 |
| Shape 3 | 0.9112 | 0.8501 | 0.8359 | 0.3711 | 3.482 | 3.6932 | 3.4116 | 2.8181 |

### 4.2. Frequency Domain Characteristics

The maximum fluctuating pressure on the bottom of the plunge pool was a key feature when studying the frequency domain characteristics. The power spectral density function usually represented the frequency characteristics of fluctuating pressure and reflected the distribution characteristics of the fluctuating energy of water flow in the frequency domain.

Figure 12 depicts the relationship between the power spectrum and frequency of different overflow shapes at 2.20 m upstream water head with different sediment concentrations, where the ordinate is the normalized power spectrum of the fluctuating pressure at a typical measuring point and the abscissa is the frequency of pressure. The results showed that the turbulent energy was most concentrated at a frequency band of 0–0.2 Hz. The power spectral density distribution curve decreased with the shape change. As the outlet became narrower, the power spectral density decreased more slowly. The variation in power spectral density was closely related to the vortex in the plunge pool. The dominant frequency of the large-scale vortex was low. The fluctuating energy was mainly concentrated in the low-frequency range and the frequency band of fluctuating energy widened as the sediment concentration increased. When the outlet became narrower, the vortex scale was wide and the spectral density increased due to the influence of air entrainment.

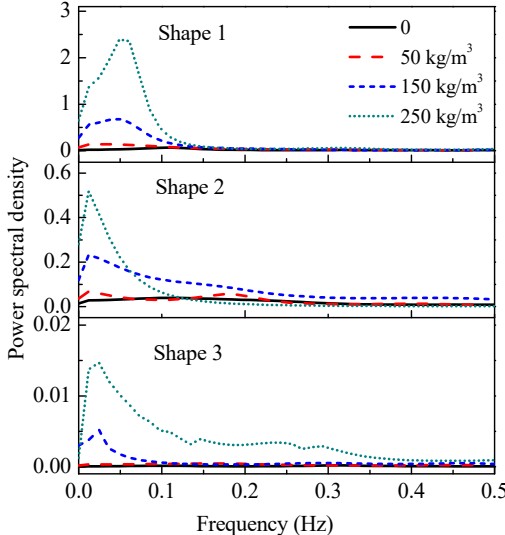

**Figure 12.** Relationship between power spectral density and frequency at 2.20 m upstream with different sediment concentrations.

The frequency decreased as the outlet shrinkage increased, and the turbulent energy was transmitted to the vortex at a low frequency and large scale. With the increase in sediment concentration, the influence of air entrainment became smaller, while the influence of viscosity increased, and the large-scale vortex of various body shapes was more concentrated. These results are similar to those of Wang and Qian [16], who conducted experiments using plastic granules to study turbulent structures flowing in open channels.

### 4.3. Time Domain Characteristics

The spatiotemporal correlation coefficient was used to characterize the interdependence of different spatial points at different time fluctuating pressures, reflecting that large-scale vortices kept their scales flowing downstream in a certain spatial range.

Figure 13 shows how the autocorrelation coefficients of various shapes were compared with fresh water and sediment-laden flow. The variation rules of fluctuating pressure correlation curves were similar for different overflow shapes. The autocorrelation curves all showed a downward trend from 1 to 0, and the autocorrelation coefficient decreased with the increase in the time interval. At time intervals of 0–5 s, the autocorrelation coefficient decreased rapidly. With an increase in time, the autocorrelation coefficient slowly decreased and gradually approached zero. As the contraction ratio decreased, the autocorrelation coefficient distribution dropped rapidly. With an increased sediment concentration, the effect of shape change on the scale of vortex was reduced. The autocorrelation coefficient decreased more slowly with the increase in sediment concentration in a given overflow shape. When the sediment concentration was 250 kg/m$^3$, the autocorrelation coefficient decreased most obviously as the outlet size shrank. These results correspond well with those in [35,36]; the longitudinal sizes of the macroscale eddies increase with the increase in sediment concentration and are mainly decided by the low-frequency, large eddies.

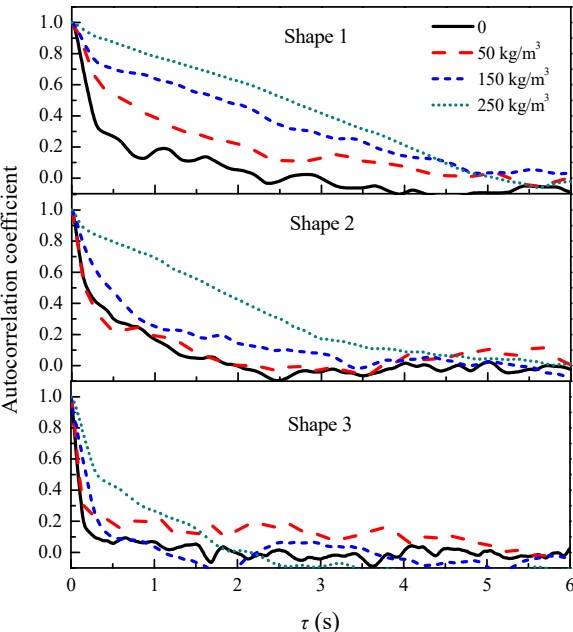

**Figure 13.** Profile of autocorrelation coefficient for various shapes in fresh water and sediment-laden flow at 2.20 m upstream with different sediment concentrations.

Due to the change in turbulent structure and surface tension with increased sediment concentrations, the scale of the vortex enlarged. The scale of vortex in the hyper-concentrated flow was larger than that of fresh water. This was also the result of the combined effects of gravity caused by sediment-laden water and air entrainment caused by overflow weir shape changes.

*4.4. Comparative Analysis*

In practical projects, a method of outlet shrinkage should be employed, such as a flaring gate pier. Li [37,38] studied the flow discharge with contraction of jet and water nappe characteristics in a high arch dam by using a flaring gate pier. The research results indicated, that due to the shrinkage of the cross section, the direction of water particles changed at different depths vertically along the exit cross section, and the water nappe was obviously made longer, with dispersion characteristics in the longitudinal direction. The discharge capacity was affected by the contraction ratio, and with the increase of water head on the weir, the effect of outlet shrinkage on the discharge capacity was more obvious. Gu [39] analyzed the characteristics of hydrodynamic load under the X-shaped flaring gate pier in a short stilling basin. The maximum fluctuating pressure increased with the widening of the outlet at the bottom of the stilling pool. Previous studies [37–39] stated the influence of the flaring gate pier on the flow regime of nappe, discharge capacity, and fluctuating pressure on the bottom of the pool. Although different objects were studied, and different experimental details were produced, the effect of outlet shrinkage of the overflowing building in the literature correspond well with those results. Sang [40] used a large-scale model to test the same shape of bucket with fresh water. Two physical models with different scales produced consistent results when comparing sediment-laden flow with fresh water. The results pointed out that the discharge capacity gradually decreased as the contraction ratio decreased in a relatively high water head of a weir, and that outlet shrinkage had little effect on the position of the pressure peak in the plunge pool.

In terms of numerical simulations, Xu [41] studied the typical three-dimensional flow pattern in the plunge pool and energy dissipation of an arch dam. The research results revealed that an energy transition occurred mainly in the region near the jet axis, while only a little water mass in the plunge pool took part in the energy dissipation. The distributions of turbulent energy and its dissipation rate were revealed, and energy dissipation was more difficult with concentrated jets. The way to optimize the plunge pool and dam outlets should be to expand the range of the shear dissipation region; the outlet shrinkage used in the literature can disperse the water nappe and accelerate energy dissipation.

The flow fluctuating load was caused by various vortices. The fluctuating pressure was the exact reflection of different scale vortices on the solid boundary. Liao [42] pointed out that a large-scale vortex component with low frequency was present in the bottom of the plunge pool. Huang [43] investigated the characteristics of pressure fluctuations in the three-phase flow of water, air, and sand based on experimental data. Air entrainment and sediment had finite effects on the distribution of the probability density function of the fluctuating pressure. This was roughly consistent with the normal distribution, as in fresh water. According to a spectrum analysis of experimental data, the fluctuating energy of high-velocity sediment-laden and aerated flow was widely distributed in frequency. The power spectrum of fluctuating pressure was basically the same as that of fresh water when the sediment concentration was lower than 100 kg/m$^3$. Gou [21] and Zhang [26] conducted model tests with a hyper-concentrated flow (sediment concentrations of 250 kg/m$^3$ or 400 kg/m$^3$). The results of [21,26,43] revealed that large eddies existed in the sediment-laden flow and conformed to the research result. However, the effect of outlet shrinkage was also considered in the research. The outlet shrinkage decreased the dominant frequency at a hyper-concentrated flow, and the large-scale vortex increased. When the sediment concentration was low, outlet shrinkage had little effect.

## 5. Conclusions

The effect of sediment concentration on energy dissipation is an urgent problem for high dams in hyper-concentrated sediment-laden rivers. Thirty-six experiments were conducted using overflows with three contraction ratios and four sediment concentrations (0 kg/m$^3$, 50 kg/m$^3$, 150 kg/m$^3$, and 250 kg/m$^3$) under three upstream water heads (2.10 m, 2.15 m, and 2.20 m). The experimental results were used to study the hydraulic characteristics of energy dissipation in terms of flow discharge, jet properties, and hydrodynamic pressure. The main conclusions were as follows:

(1) The discharge flow was related to the upstream water head, regardless of the sediment concentration. The flow coefficient of different overflow shapes generally first increased and then decreased with an increase of upstream water head. When the water head of weir was below 10 cm, the discharge capacity was basically not affected by the width of the outlet. With an increase of water head, the influence of outlet shrinkage on the discharge capacity gradually increased. As the contraction ratio decreased, the water depth became greater, affecting the discharge capacity, and the flow coefficient decreased accordingly.

(2) The effect of outlet shrinkage on flow regime decreased with increasing sediment concentration. In freshwater and sediment-laden flow (50 kg/m$^3$), the water nappe became narrower with outlet shrinkage, air entrainment increased, and the length of the jet trajectory increased. In a sediment-laden flow (150 kg/m$^3$ or 250 kg/m$^3$), it can be seen from the completeness of the nappe that the length of the jet was reduced because the air entrainment was significantly reduced and the influence of gravity gradually increased. As the sediment concentration increased, the diffusion of the water nappe decreased, and the effect of the outlet shape on the nappe reduced.

(3) Fluctuating pressure is the result of the movement and interaction of vortices of different scales in the pool. The time-averaged pressure and fluctuation pressure both exhibited peaks, describing the impact of the jet on the bottom of the plunge pool. As the width of the outlet became narrower, the air entrainment of the water nappe increased, and the relative time-averaged pressure on the bottom of the plunge pool dropped. The difference between the maximum RMS is similar to the different overflow weir shapes with different sediment concentrations. The shape change causes a change in fluctuating pressure, and as the shape of the outlet becomes narrower, the fluctuating pressure decreases.

(4) When considering hydrodynamic pressure, the main frequency of the fluctuating pressure was from 0 to 0.2 Hz. The outlet shrinkage caused the frequency range to expand. The dominant frequency became lower with the increasing sediment concentration. The variation of the autocorrelation coefficient distribution was similar to the power spectral density. With increased sediment concentration, the effect of the overflow shape on the vortex scale was reduced and the autocorrelation coefficient distribution decreased more slowly.

The findings of this study provide invaluable insights into the effects of overflow shape on energy dissipation in high dams on sediment-laden rivers. The behaviors and relationships determined in this study establish a foundation for the informed design of dissipator structures such as plunge pools, improving the operational safety of the dam.

**Author Contributions:** All authors contributed to the research work. J.L. and F.L. conceived and designed the experiments; H.Y. and H.L. performed the tests; H.Y. analyzed the data; and H.Y. and W.G. drafted the paper. All authors have read and agreed to the published version of the manuscript.

**Funding:** This research was funded by the National Key R&D Program of China (Grant Number 2016YFC0401704); National Natural Science Foundation of China (Grant Number U1765202); Science Fund for Creative Research Groups of the National Natural Science Foundation of China (Grant Number 51621092); and Science and Technology Project of China Huaneng Group (Grant Number HNKJ15-H12).

**Conflicts of Interest:** The authors declare no conflict of interest.

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
