# Peer review of "Effects of Outlet Shrinkage on Hydraulics in Hyper-Concentrated Sediment-Laden Flow"

_applsci, doi:10.3390/app10041332_

Round 1
Reviewer 1 Report
Review #:
#1: The authors study the effects of outlet shrinkage on hydraulics in hyper-concentrated sediment-laden flow. The results have practical value to support researchers in the field. Although the topic is of interest, it requires significant amendments before being published.
#2: The abstract is poorly written, and it did not convey any message to the reader. Therefore, the authors are required to re-write a convincing abstract that is clear and concise.
#3: The entire writing style and English grammar need crucial modification because it makes the paper difficult to articulate.
#4: The authors employed an experimental method in their study without calibrating their results with an existing field case or numerical simulation. There is a critical need for both mentioned approaches to justify their experimental outcome. I suggest the authors employ those methods for validations/verification of their results.
Author Response
Dear reviewer,
We sincerely appreciate your review. We studied the all comments carefully, and tried our best to revise the manuscript. The respond was uploaded to the attachment.

Reviewer 2 Report
REVIEW OF MANUSCRIPT applsci-702704
TITLE: Effects of outlet shrinkage on hydraulics in 3 hyper-concentrated sediment-laden flow
The reviewer raises the following comments regarding the manuscript:
Comment 1 (Originality-literature review)
The manuscript provides a very good literature review in the field, and the authors state clearly the added value of the research in comparison with previous works in the field and in comparison with earlier research of the authors’ team.
Comment 2 (Methodology)
2.1 Mathematical system: The basic mathematical background of the method is included. The Nyquist law has to be included as well (no matter how familiar it is, a basic description is necessary).
2.2 Numerical results: The paper provides typical results, figures and tables for experiments and simulations. The data of the results, as well as the discussion of them, present clearly the information the authors want to transfer to the reader.
2.3 There is no validation procedure: How do the authors demonstrate the validity of this proposed method and how do they evaluate the final results?
Comment 3 (English language and specific changes)
3.1 There is some room for English language improvement. For example in phrase, page 1, line 32: “ At present, two types of energy dissipation methods were often used: one was traditional energy dissipation (hydraulic jump energy dissipation, surface flow energy dissipation and ski-jump energy dissipation); the other was new energy dissipation. ..” there are grammatically errors. The whole manuscript has to be checked.
3.2 The “DASP” definition has to be included. (Page 4, line 107).
Author Response

(The authors gave the same response as above.)

Round 2
Reviewer 1 Report
The authors have addressed must of the issues raised on their early submission, but there is a need to improve the writing style. For example, the first sentence in the abstract does not provide a reader with any information "The appropriately shape of the releasing building is the important design on energy dissipation for a high dam on the sediment-laden river." Such a sentence and many others in the manuscript require restructuring before acceptance.